# Information content of soil hydrology in a West Amazon watershed as informed by GRACE

Elias C. Massoud[1,2]*, A. Anthony Bloom[1], Marcos Longo[1], John T. Reager[1], Paul A. Levine[1], John R. Worden[1]

[1]Jet Propulsion Laboratory, California Institute of Technology, Pasadena, CA, USA
[2]Department of Environmental Science, Policy, and Management, University of California, Berkeley, Berkeley, CA, USA

*Correspondence to*: Elias C. Massoud (eliasmassoud@berkeley.edu)

**Abstract.** The seasonal-to-decadal terrestrial water balance on river basin scales depends on several well-characterized but uncertain soil physical processes, including soil moisture, plant available water, rooting depth, and recharge to lower soil layers. Reducing uncertainties in these quantities using observations is a key step towards improving the data fidelity and skill of land surface models. In this study, we quantitatively characterize the capability of Gravity Recovery and Climate Experiment (NASA-GRACE) measurements —a key constraint on Total Water Storage (TWS) —to inform and constrain these processes. We use a reduced complexity physically based model capable of simulating the hydrologic cycle, and we apply Bayesian inference on the model parameters using a Markov Chain Monte Carlo (MCMC) algorithm, to minimize mismatches between model simulated and GRACE-observed TWS anomalies. Based on the prior and posterior model parameter distributions, we further quantify information gain with regards to terrestrial water states, associated fluxes, and time-invariant process parameters. We show that the data-constrained terrestrial water storage model can capture basic physics of the hydrologic cycle for a watershed in the western Amazon during the period of January 2003 through December 2012, with an $r^2$ of 0.98 and RMSE of 30.99 mm between observed and simulated TWS. Furthermore, we show a reduction of uncertainty in many of the parameters and state variables, ranging from a 2% reduction in uncertainty for the porosity parameter to an 85% reduction for the rooting depth parameter. The annual and interannual variability of the system are also simulated accurately, with the model simulations capturing the impacts of the 2005-2006 and 2010-2011 South America droughts. The results shown here suggest the potential of using gravimetric observations of TWS to identify and constrain key parameters in soil hydrologic models.

## 1 Introduction

The terrestrial water balance depends on many physical processes, including soil moisture, plant available water, rooting depth, recharge to lower soil layers, among others, and these processes depend on each other in a dynamical way (Margulis et al., 2006; Massoud et al., 2019a, 2020a). Some variables, such as precipitation, surface runoff, or soil moisture, can be directly observed in the field or by airborne measurements (Walker et al., 2004; Swenson et al. 2006; Durand et al., 2009; Liu et al.,

2019), but other processes, such as evapotranspiration or groundwater storage changes, are more difficult to detect and observe (Tapley et al., 2004; Pascolini-Campbell et al., 2020). Model simulations are one tool that can be used to fill gaps where our understanding of the hydrologic cycle is incomplete or missing (Purdy et al., 2018; Massoud et al., 2018a). Different types of models exist, such as distributed models with dozens or hundreds of parameters that simulate process-based physics at the grid

scale but are extremely expensive to run (Vivoni et al., 2007; Hanson et al., 2012; Longo et al., 2019; Massoud et al., 2019b), or lumped models that aggregate information in space and time to reduce the cost of model simulations while maintaining accuracy when compared to measurements (Manfreda et al., 2018; Massoud et al., 2018b). Recent advances in model-data fusion have paved the way to merge land model simulations with observations (Girotto et al., 2016; Khaki et al., 2017, 2018; Quetin et al. 2020; Sawada 2020), limiting the need for process representation in the model and increasing the efficiency in

the inference of unknown physical processes, such as hydrologic variables that cannot be directly measured.

The wealth of data available today, including in-situ measurements, flux towers, or satellite data from remote sensing, has made it increasingly possible to fuse model simulations with observations. This has been shown in several works in the literature so far (Massoud et al., 2018ab; Seo and Lee 2020). One set of satellite observations that has been very popular in the literature is the NASA Gravity Recovery and Climate Experiment (GRACE) pair of satellites (Tapley et al., 2004). Satellite

observations of Earth's gravity field from GRACE are processed routinely into estimates of surface mass change and can provide information about basin-scale dynamics of hydrologic processes. GRACE mass change estimates can be combined with other hydrologic information, such as model simulations or in situ observations, to infer hydrologic parameters and state variables (Famiglietti et al., 2011; Xiao et al., 2017; Trautmann et al., 2018; Massoud et al., 2018a, 2020a; Liu et al., 2019). Numerous studies in the literature have assimilated information from GRACE into models for a better understanding of how

groundwater systems behave on different scales (Zaitchik et al., 2008; Houburg et al., 2012; Reager et al., 2015).

Across a variety of climate and land surface models (Christoffersen et al., 2016; Purdy et al., 2018; Massoud et al., 2019a; Schmidt-Walter et al., 2020), hydrology process parameters— both physical states and empirical process variables— constitute a major uncertainty in models. Uncertain variables include rooting depth, infiltration rates, water retention curves, among other soil physical processes, which are governing factors in the dynamic evolution of soil water states. Typically,

models prescribe these parameters either by default values or by calibrating the models in well studied and extensively measured domains. However, few efforts have been made to assess uncertainties tied to the choice of these parameter values. Many of these prescribed parameters come from observational studies, such as Hodnett and Tomasella (2002) and the ones indicated in Marthews et al., (2014). Studies such as these optimize parameters, along with their dependence on soil characteristics, to represent field measurements of water retention curves. However, the samples are often restricted to few

sites and not necessarily representative of larger regions. Furthermore, the models may have limitations in their physical process representation, which could induce bias in predictions if these parameters as used as the "truth". In general, information on parameters can be inferred with high confidence using datasets obtained from remote sensing.

In this study, we demonstrate the ability of the decadal GRACE Total Water Storage (TWS) record to inform and reduce uncertainties of terrestrial hydrologic processes regulating the seasonal and inter-annual variability of TWS in the

western Amazon, the Gavião watershed, for the period January 2003 through December 2012. To achieve this, we use a model of necessary complexity to represent the first order controls on seasonal-to-decadal soil moisture dynamics, including soil moisture, soil water potential, plant available water and rooting depth. To characterize and quantify information content of the GRACE record, we employ a Bayesian model-data fusion approach to constrain model parameters (namely initial states and time-invariant process variables), such that differences between GRACE and simulated TWS anomalies are statistically minimized. We henceforth collectively refer to time-invariant parameters governing soil moisture states—such as porosity, rooting depth and hydraulic conductivity coefficients—as model process parameters throughout the manuscript.

Our study is set up as follows: We describe in section 2 the TWS model, the GRACE TWS data used to constrain our simulations, and the Bayesian method used to infer the model parameters. In section 3, we define the model's physically based equations, introduces the time-invariant model parameters that are optimized and inferred, and highlights our findings and results. We summarize our work in section 4 and discuss the implications of our results and priority points for further developments.

## 2 Data and Methods

### 2.1 Data-constrained Terrestrial Water Storage model

We employ a model of necessary complexity to represent basin scale hydrologic processes that regulate the storage and movement of water on monthly timescales, as shown in Figure 1. The model includes two soil layers, where the top layer represents the water that is available to plants via roots (Plant Available Water, or PAW), and the bottom layer representing depths of the soil that plant roots cannot access (Plant Unavailable Water, or PUW). The model uses monthly time steps to integrate the state variables and is driven with hydrologic flux variables such as evapotranspiration and precipitation. The model also includes other processes such as infiltration into the soil, surface runoff, drainage from each layer, recharge into the lower soil layer, and various model parameters (listed in Table 1) that control the simulations.

The model includes 13 parameters that represent process-based hydrologic mechanisms, ones that are hypothesized to be influential on basin-scale monthly resolution model simulations of the hydrologic cycle. As depicted in Figure 1, there are two soil layers representing the PAW and PUW pools. Each of the two separate soil layers has its own inferred physical properties, such as the depth of each layer, soil moisture initialization, porosity, field capacity, and retention capabilities. Various fluxes are represented in the model, such as precipitation (P), evapotranspiration (ET), infiltration, surface runoff, and drainage. The parameters of the model dictate the simulation of each process in the hydrologic cycle, and by adding the two water pools (PAW + PUW), an estimate of total water storage (TWS) can be generated which can then ultimately be compared with the GRACE-based TWS.

We describe here the model equations that dictate how the TWS is calculated in the model. To start, we know from the water mass continuity that the changes in TWS in the soil is the equivalent to the balance between input (precipitation, P)

and outputs (evapotranspiration ET, and the total loss through drainage and runoff Q). In effect, P and ET are prescribed boundary conditions for the model. In this version of the model,

$$TWS_t = M_t^{PAW} + M_t^{PUW} \tag{1}$$

where $M_t^{PAW}$ represents the plant available water and $M_t^{PUW}$ is the plant unavailable water at each month, $t$. The soil is represented this way in the model because plants cannot access all the water stored in the ground, therefore two separate layers are used to represent the soil water in the rooting zone (PAW) and the soil water that is not accessible to plants (PUW).

The following model equations are used to represent the storage and flow of water in the model. The mass continuity equations for water stored in the $M^{PAW}$ and $M^{PUW}$ layers are:

$$M_{t+1}^{PAW} = M_t^{PAW} + I_t - D_{t,PAW} - F_t - ET_t \tag{2}$$

$$M_{t+1}^{PUW} = M_t^{PUW} - D_{t,PUW} + F_t \tag{3}$$

where $I_t$ is the infiltration into the top soil layer, $D_{t,PAW}$ and $D_{t,PUW}$ are the drainage terms for each layer, $F_t$ is the recharge in between layers, and $ET_t$ is the evapotranspiration term at each month, $t$. We assume that a fraction of precipitation cannot infiltrate in the soil. This occurs because during rainy events, the precipitation rates often exceed the percolation rates of the near-surface soil, which may become temporarily saturated. These processes occur at sub-monthly scales and cannot be explicitly accounted for in the model; therefore, we use a phenomenological approach that assumes a maximum infiltration rate:

$$I_t = I_{max} \left( 1 - e^{\frac{-P_t}{Imax}} \right) \tag{4}$$

where $P_t$ represents the precipitation rate at each month and $I_{max}$ is the parameter that represents the maximum infiltration. The excess precipitation is lost as surface runoff ($S_t$) and never enters in the soil storage:

$$S_t = P_t - I_t \tag{5}$$

The recharge flux between the PAW and PUW layers ($F_t$, positive when the flow goes from PAW to PUW) can be defined by the Darcy's law, relating the difference in potentials between the two layers:

$$F_t = \rho_\ell K_{t,layer} \left[ \frac{10^{-6}}{\rho_\ell g} \frac{\Psi_{t,PAW} - \Psi_{t,PUW}}{\frac{1}{2}(L_{PUW} - L_{PAW})} + 1 \right] \tag{6}$$

where $\Psi_{t,PAW}$ and $\Psi_{t,PUW}$ [MPa] are the soil matric potential of each layer at each month, $\rho_\ell = 1000$ kg m$^{-3}$ is the water density, $g = 9.807$ m s$^{-2}$ is the gravity acceleration, $K_{t,layer}$ [m s$^{-1}$] is the hydraulic conductivity of the source layer (i.e. PAW if $F_i$ is positive, and PUW if $F_i$ is negative), and $L_{PUW}$ (rooting depth) and $L_{PAW}$ (remainder of soil depth) are the parameters that represent the thickness of each layers [m].

Then, the soil matric potential of each layer is defined as a function of relative soil moisture ($SM_{t,layer}$), following Brooks and Corey (1964):

$$\psi_{t,layer} = \psi_{porosity} \left(\frac{1}{SM_{t,layer}}\right)^b \tag{7}$$

where $\psi_{porosity} = -0.117$ MPa, and the parameter $b$ corresponds to the inverse of the pore size distribution index (Marthews et al. 2014). The unsaturated hydraulic conductivity ($K_{t,layer}$) is defined following Campbell (1974):

$$K_{t,layer} = K_0\, SM_{t,layer}^{2b+3} \tag{8}$$

where $K_0$ [m s$^{-1}$] is the parameter that represents the saturated hydraulic conductivity, and the parameter $b$ is the same as in
Eq. (7). The drainage function is parameterized as the removal of water that exceeds the field capacity, to represent fast (sub-monthly) loss of water under near-saturated conditions:

$$D_{t,layer} = \frac{\max\left(0, \psi_{t,layer} - \psi_{field}\right)}{Q_{excess}\left(\psi_{porosity} - \psi_{field}\right)} \tag{9}$$

where the scaling term $Q_{excess}$ is a free parameter from 0-1 that removes a fraction of SM excess above field capacity, $\psi_{field}$.

Lastly, one thing to note is that precipitation and ET biases in the Amazon are known to be significant, and ET can
even have an inverted seasonal cycle. The model is capable of substantially relaxing and constraining the simulated evapotranspiration ($ET_t$) and precipitation ($P_t$) values at each month, through the parameterization and inference of scale factors ($P_{scale}$ and $ET_{scale}$). The data set used for P at each month, namely $P_{data,t}$, is derived from precipitation measurements from the Tropical Rainfall Measuring Mission (TRMM) 3B42 (Huffman et al., 2007), provided at 0.25° × 0.25° and 3-hourly spatiotemporal resolutions. The data sets used for ET at each month, namely $ET_{data,t}$, is derived following the approach in
Swann and Koven (2017) and Shi et al., (2019). That is, monthly total ET is derived from satellite observations of precipitation and TWS and ground-based measurements of river runoff. Unlike the ET retrievals from the Moderate Resolution Imaging Spectroradiometer, which have been shown to be seasonally biased in the wet tropics (Maeda et al., 2017; Swann and Koven, 2017), this ET estimation is robust across seasons (Swann and Koven, 2017). Runoff data sets for each watershed are obtained from the Observation Service for the geodynamical, hydrological, and biogeochemical control of erosion/alteration and
material transport in the Amazon (SO-HYBAM) in situ river gauge discharge measurements (discharge measurements can be found at http://www.ore-hybam.org/). With these three data sets, we estimate subbasin-based monthly ET.

To clarify this more, there are 3 different derivations used for the TWS variable. These 3 estimates provide a sense of uncertainty for the TWS. The uncertainty from the GRACE product is used in the likelihood function of the MCMC algorithm when fitting the model simulated TWS to the GRACE derived TWS. Then, there is also 3 products used in the
150 precipitation and the runoff driving variables that were used, to get a sense of the uncertainty in each variable. To estimate the ET driving variable in this work, we use the mean of the TWS, P, and Q products and create a water balance that will allow us

to estimate a mean for the ET driving variable. Then by application of the ET scaling parameter, we try to estimate whether our initial calculation of ET required any scaling to match the data. Therefore, even though the GRACE TWS is somehow used in the derivation of the ET data, the uncertainty that is applied throughout the work allows us to still estimate ET that is not dependent on the GRACE data. See Shi et al., (2019) for more details on this derivation. In essence, the simulated fluxes are represented as $ET_t = ET_{scale} * ET_{data,t}$ for evapotranspiration, and $P_t = P_{scale} * P_{data,t}$ for precipitation, where $ET_{scale}$ and $P_{scale}$ are inferable parameters. Combining all these equations in the logical flow presented in Figure 1 of the manuscript allows the model to simulate total water storage as $TWS_t = M_t^{PUW} + M_t^{PAW}$. This model has not been presented before in previous literature, and this manuscript is the first to report on the model simulation results.

The parameters of the model will be inferred such that the TWS in the model simulations match the observed GRACE TWS data. Since GRACE TWS is known to have the smallest uncertainties in the water budget (c.f. Pascolini-Campbell et al., 2020), we use this information to infer and understand the more poorly constrained variables or processes in the model. In this case study, we use the model for the Gavião Watershed, located in the western Amazon (for location of watershed refer to the map in Figure 1). We chose the Gavião Watershed for this study due to sufficient data availability, and because there is a strong seasonal cycle for this watershed, which allows the model to capture hydrologic signals more efficiently during the parameter inference.

### 2.2 GRACE Data for Total Water Storage (TWS)

NASA's GRACE mission (Tapley et al., 2004) has proven to be an extremely valuable tool for regional to global scale water cycle studies (Famiglietti 2014; Reager et al., 2015; Massoud et al., 2018a, 2020a). GRACE data have been widely used to diagnose patterns of hydrological variability (Seo et al., 2010; Rodell et al., 2009; Ramillien et al., 2006; Feng et al., 2013), to validate and improve model simulations (Döll et al., 2014; Güntner, 2008; Werth and Güntner, 2010; Chen et al., 2017; Eicker et al., 2014; Girotto et al., 2016; Schellekens et al., 2017), to constrain decadal predictions of groundwater storage (Massoud et al., 2018a), and to enhance our understanding of the water cycle on regional to global scales (Syed et al., 2009; Felfelani et al., 2017; Massoud et al., 2020a). Total Water Storage (TWS) estimates from GRACE include all of the snow, ice, surface water, soil water, canopy water, and groundwater in a region, and when combined with auxiliary hydrologic datasets, TWS can be utilized to infer process information on model parameters or other model states.

Various recent studies have demonstrated that GRACE derived estimates of variations of TWS can provide freshwater availability estimates with sufficient accuracy (Yeh et al., 2006; Zaitchik et al., 2008; Massoud et al., 2018a). These GRACE based methods have been applied to regions such as Northern India (Rodell et al., 2009; Tiwari et al., 2009), the Middle East (Voss et al., 2013; Forootan et al., 2014; Massoud et al., 2021), Northern China (Moiwo et al., 2009; Feng et al., 2013), California (Famiglietti et al., 2011; Scanlon et al., 2012; Xiao et al., 2017; Massoud et al., 2018a, 2020a), Northern mid- to high latitudes (Trautmann et al., 2018), and the Amazon (Swann and Koven 2017), among others. In this study, estimates of TWS are obtained from the GRACE retrievals of equivalent water thickness (Landerer & Swenson, 2012; Sakumura et al., 2014; Wiese et al., 2016). We use three GRACE TWS retrievals from the spherical harmonic data versions generated by the

Center for Space Research (CSR), GeoforschungsZentrum Potsdam (GFZ), and Jet Propulsion Laboratory (JPL). These three GRACE TWS retrievals are 1-degree solutions of land field products (each was downloaded from ftp://podaac-ftp.jpl.nasa.gov/allData/tellus/L3/land_mass/RL05/). We calculate the arithmetic mean of the three GRACE TWS retrievals to represent TWS used in Eq. (1). We used this GRACE product to constrain simulations of the hydrologic model described in Section 2.1 for the Gavião watershed from January 2003 through December 2012.

### 2.3 Bayesian Parameter Inference with MCMC

In this study, we aim to estimate parameters of a medium complexity model that simulates the hydrologic cycle using physics-based equations that capture large scale dynamics of the watershed. We showcase how the data-constrained physically based model can simulate the hydrologic cycle by fusing the model with auxiliary observations. When simulated on its own, the model can represent a wide range of physical possibilities, but when calibrated and trained to fit some desired observed metric, the model simulations begin to represent the underlying physical system it is being trained to. Many tools exist to achieve model-data fusion, such as Bayesian parameter inference with Markov Chain Monte Carlo (MCMC) algorithms (Schoups and Vrugt, 2010; Bloom et al., 2015; Vrugt 2016; Vrugt and Massoud 2018; Massoud et al., 2019c, 2020b) or data assimilation (Reichle et al., 2002; Vrugt et al. 2005; Girotto et al., 2016; Khaki et al., 2017, 2018; Massoud et al., 2018b). These state-of-the-art tools require enough computational cost but can ensure that the underlying system dynamics are being accurately replicated to an agreeable amount of uncertainty. The model parameters in this study are estimated using Bayesian inference with MCMC (Vrugt and Massoud 2018), where the final estimated distributions are not required to follow any form, such as Gaussian or bimodal, amongst others. The final estimates of the model parameters, shown later to be the posterior of $\theta$ in Eq. (12), are the posterior solutions and are utilized to constrain the spread of uncertainty in the simulations.

In recent decades, Bayesian inference has emerged as a working paradigm for modern probability theory, parameter and state estimation, model selection and hypothesis testing (Vrugt and Massoud 2018). According to Bayes' theorem, the posterior parameter distributions, $P(A|B)$, depend upon the prior distributions, $P(A)$, which captures our initial beliefs about the values of the model parameters, and a likelihood function, $L(\theta)$, which quantifies the confidence in the model parameters, $\theta$, considering the observed data, $\mathbf{Y}$. The likelihood function is a critical property of this calculation. This section shows the derivation of the likelihood function used in this study. According to Bayes' Theorem, the probability of an event is estimated based on prior knowledge of conditions that might be related to the event. In equation form, this looks like:

$$P(A|B) = \left( \frac{P(B|A) * P(A)}{P(B)} \right).$$  (10)

For the purposes of this study, we can express P(A) as the prior information of our calculation, which assumes log-uniform distribution for all parameters and the probability outside the parameter bounds is equal to 0 (the minimum and maximum values for each parameter are reported in Table 1). P(B) is the evidence and is a normalizing constant and therefore taken out of the equation. This leaves us with:

$$P(A|B) \propto P(B|A) \; ; \; P(A|B) \propto L(\boldsymbol{\theta}). \tag{11}$$

where $P(A|B)$ is the final distribution of the model parameters, or the posterior of $\boldsymbol{\theta}$ in Eq. (12) described in the next paragraph, and $P(B|A)$ is equivalent to the chosen likelihood function, $L(\boldsymbol{\theta})$, also described in the next paragraph. Therefore, the MCMC algorithm samples model parameter combinations ($\boldsymbol{\theta}$) that will maximize the fit to the GRACE data, and thus will maximize the value of the likelihood function, $L(\boldsymbol{\theta})$.

The observed data in this case study is the GRACE satellite observations, and our goal is to find the optimal set of model parameters, $\boldsymbol{\theta}$, that produces a model simulation, $\mathbf{X}(\boldsymbol{\theta})$, which maximizes the fit, or the likelihood, relative the observations. Our likelihood function is therefore set up as:

$$L(\boldsymbol{\theta}) = -\frac{1}{2\sigma_{\text{GRACE}}^2}\sum_t \left[ \mathbf{Y}_{\text{GRACE},t} - \mathbf{X}_{\text{Model},t}(\boldsymbol{\theta}) \right]^2 \tag{12}$$

where $t$ refers to the time index (in months) of the simulations, $\mathbf{Y}_{\text{GRACE},t}$ is the observed GRACE data at month $t$, $\mathbf{X}_{\text{Model},t}(\boldsymbol{\theta})$ is the optimized model simulations at month $t$ using the parameters $\boldsymbol{\theta}$, and $\sigma_{\text{GRACE}}^2$ is the uncertainty associated with the GRACE data which was chosen to be a homogeneous 50 mm/month for our applications. Since GRACE data is represented as anomalies from climatology, we format the model simulations into anomalies as well to perform this model-data fitting experiment. That is:

$$\mathbf{Y}_{\text{GRACE},t} = \mathbf{TWS}_{\text{GRACE},t} - \text{mean}(\mathbf{TWS}_{\text{GRACE}}) \tag{13}$$

indicating that the form of the GRACE observations is in climatological anomalies. Furthermore, we format the model simulations in this manner for the parameter inference, as follows:

$$\mathbf{X}_{\text{Model},t} = \mathbf{TWS}_{\text{Model},t} - \text{mean}(\mathbf{TWS}_{\text{Model}}) \tag{14}$$

We apply Bayesian inference on the model parameters in an optimization framework and sample the likelihood function in Eq. (12). This allows for the inference of the model parameters, or $\boldsymbol{\theta}$. These inferred model parameters will be used to inform and constrain the spread of uncertainty in the model simulations.

Successful use of the MCMC application in a Bayesian framework depends on many input factors, such as the number of chains, the prior used for the parameters, number of generations to sample, the convergence criteria, among other things. For our application, we use the adaptive Metropolis-Hastings MCMC, as described in Bloom et al., (2020). We use C=4 chains, the prior was a log-uniform distribution for each parameter and the ranges shown are listed in Table 1, the number of generations was set at G=100,000, and the convergence of the chains relied on the Gelman and Rubin (1992) diagnostic, where we applied the commonly used convergence threshold of R=1.2. Given the high efficiency of running this parsimonious model (as compared with other high dimensional and expensive models), it was computationally feasible to obtain the set of G=100,000 simulations for the MCMC algorithm (i.e., less than one hour of CPU time to perform the parameter inference).

### 2.4 Averaging Kernel Matrix

To better quantify the reduction of uncertainty for each parameter, we apply an Averaging Kernel (**AK**) calculation (Worden et al., 2004), which is typically a measure of how a modelled state (posterior) is sensitive to changes in the "true" state (prior) and is a method that is common for satellite retrievals. The **AK** matrix is calculated as follows:

$$\mathbf{AK} = \mathbf{I} - \frac{\text{cov}(\mathbf{Posterior})}{\text{cov}(\mathbf{Prior})} \tag{15}$$

where **AK** is the diagonal vector of the averaging kernel matrix, **I** is the identity matrix, **Posterior** is the Bayesian parameter posteriors sampled with MCMC, **Prior** are samples randomly drawn from the prior distribution, and cov is the covariance function. We take the main diagonal of the **AK** matrix, which represents uncertainty reduction from the prior to the posterior parameter distributions. The **AK** diagonal values for each parameter are listed in Table 1 under 'AK Diagonal'. A value of **AK** = 1 represents a 100% reduction in uncertainty, while a value of **AK** = 0 represents no information gain and therefore no reduction in uncertainty.

## 3 Results and Discussion

### 3.1 Sensitivity of TWS variability to model parameter

To characterize the sensitivity of the monthly TWS variability to model parameters, we perturb posterior parameters and generate corresponding TWS simulations. Figure 2 shows the sensitivity of the model simulated TWS to minor perturbations in parameter values. In these plots, the green curves show changes in simulated TWS (d TWS) when each parameter is perturbed (d Par) by 1% of its prior range, indicating the magnitude and the time steps of model sensitivity. Results in these plots show that sensitivity to initial conditions is larger for the first 12-month period but are diminished after that. Furthermore, the sensitivity of simulated TWS varies between wet and dry seasons.

The rooting depth parameter (Figure 2A) is sensitive during initialization as well as during the wet periods, the maximum infiltration parameter (Figure 2B) seems to only be sensitive during the wet periods, and the parameter representing the initialization of soil moisture in the top layer (Figure 2C) is only sensitive during initialization. Figure S1 in the supplementary section shows how the remaining parameters affect TWS sensitivity. To summarize these curves in a single value (i.e. [mm change in TWS per 1%-unit change in parameter]), we show in Table 1 under 'TWS sensitivity' the aggregated value for each parameter, calculated as the mean variance of all (d TWS / d Par) curves for each parameter.

### 3.2 Posterior model parameters and simulated states

#### 3.2.1 Model parameters, TWS, and states - the Gavião watershed

We apply Bayesian inference on the model parameters and simulations and optimize the fit to the GRACE data to obtain posterior solutions of the model parameters. We apply this parameter inference for 3 basins. The first is the Gavião watershed (shown in Figure 1), which has a generally wet climate. We then perform the same parameter inference to a basin that is more

wet than Gavião and is located upstream from the Acanaui river gauge station (hereafter called Basin 1), and to a basin that is drier than Gavião and is upstream from the Guayaramerin river gauge station (hereafter called Basin 2).

For the Gavião watershed, the prior and posterior parameter distributions are shown in Figure 3, and the median value for these distributions is listed in Table 1 under 'MCMC' for each parameter. We investigated how the estimated parameter values we find in this study compare with other studies in the literature. For example, the retention parameter 'b' in our study is estimated

to be around 2, which is lower than the tabulated values of Cosby et al., (1984), Tomasella and Hodnett (1998), or Marthews et al. (2014). Of course, the model in this study is simulated at much coarser resolution, and the physical meaning of these parameters may change due to processes being solved at very different scales. This is an important message for the interpretation of these results, as taking a model developed in one scale and applying it to a different scale can induce spurious errors if parameters are not adequately constrained at the intended resolution. We found that most parameters exhibited a

significant uncertainty reduction for the Gavião watershed. To quantify this reduction of uncertainty, we apply an Averaging Kernel (**AK**) calculation. The results from the **AK** matrix are listed in Table 1 under 'AK diagonal', and they indicate that significant uncertainty reduction occurs in some parameters, namely the depth of the PAW layer (rooting depth) and depth of the PUW layer, as well as the retention and maximum infiltration parameters. In contrast, we found porosity, conductivity at saturation, and $\psi_{field}$ exhibited the smallest relative uncertainty reductions.

In Figure 4 we show the model simulations of 10-year monthly TWS for the Gavião watershed, including the prior and the posterior simulations, and compare these with the values obtained from satellite data (GRACE TWS). Posterior ranges of the model simulated TWS are shown in the orange envelopes, and precipitation values used to drive the model are shown to indicate wet vs dry periods. Results in Figure 4 show that GRACE-informed soil hydrologic model simulations (posterior) can capture the monthly TWS compared to concurrent GRACE measurements, with an $r^2=0.9837$ and RMSE=30.99 mm

between observed and simulated TWS. Comparing this result with the prior model simulations (mean of the prior shown in Figure 4, and the distribution from the prior is shown in Figure S2), we see a major improvement in the constrained posterior model simulations. The mean prior has an $r^2=0.4360$ and RMSE=410.50 mm compared to the GRACE TWS, and the range of the prior simulations in Figure S2 span a wide range of possibilities. This result indicates that this simple model can accurately simulate TWS in the Gavião watershed when the parameters are inferred using GRACE measurements as a fitting target.

The model is then simulated using all samples from the posterior, which provides posterior solutions for the state variables. These are shown in Figure 5, which displays specific model processes for the Gavião watershed (map of the basin shown in the bottom right panel of Figure 5). The matric potential of plant available water (PAW $\psi$) and the matric potential of plant unavailable water (PUW $\psi$) represent the suction pressure in each soil layer that is associated with dryness/wetness. In other words, a completely wet soil layer would have a matric potential of 0 and higher levels of dryness result in more

negative matric potential values. Based on the results in Figure 5AB, the PUW layer seems to have more wetness, and therefore less suction pressure, for this watershed (i.e., values closer to 0 for the PUW layer). The recharge value (PUW -> PAW flux) represents the flux of water from the top layer to the bottom layer, where negative values indicate a downwards flux. Results

in Figure 5C show there is a continually flowing downwards flux of water from the top layer (PAW) to the bottom layer (PUW), roughly at the magnitude of 50-100 mm/month. The discharge values represent the drainage from the top layer (Q PAW) and from the bottom layer (Q PUW). The results in Figure 5DE show that there is drainage from the top layer (PAW) that peaks in the wet season at roughly 40 mm/month, and there is a drainage that follows a seasonal cycle from the bottom layer (PUW) at around 40-80 mm/month. The infiltration represents the water that infiltrates from the surface into the top soil layer. According to Figure 5F, this flux also follows a seasonal cycle, with about 250 mm/month infiltrated into the top layer during the wet season and dropping to roughly 50 mm/month in the dry season. Lastly, soil moisture of the top (SM PAW) and bottom layers (SM PUW) represent the state of soil moisture in each layer. Based on the results in Figure 5GH, the PUW layer seems to have more wetness, and therefore higher soil moisture values, for this watershed and these results correspond to what is seen for the matric potential in Figure 5AB (i.e., more wetness in the PUW layer). In Figure 5, the ranges shown in orange envelopes are the posterior ranges, indicating the range of possible solutions for each GRACE-informed state variable for the Gavião watershed. Some dynamical constraints were applied in the Bayesian optimization, such as $SM_{1,t0}$ and $SM_{2,t0}$ are greater than 0.1 but less than 0.5 [$m^3/m^3$]. The rationale for these 'common-sense' rules follows that of Bloom and Williams (2015), to ensure that non-realistic physical properties of the system are not allowed.

The resulting model simulations are largely affected by the way that ET is used in the model. We described in the methods section how ET is calculated in our study, and it is important to note that there are alternative approaches for prescribing watershed ET. For example, FLUXCOM (Jung et al., 2020), JPL-PT ET (Fisher et al., 2009) or parsimonious prognostic ET scheme (Liu et al., 2021) estimates can provide robust alternatives for the residual-based ET approach.

### 3.2.2 Interpretation of results

The posterior parameters and model simulations provide information that can be used to identify and estimate the processes responsible for TWS variability in this watershed. Insights of rooting depth (histograms in Figure 3) are critical for determining resilience of rootzone water storage during dry season events (c.f. Lewis et al., 2011, Shi et al., 2019, Liu et al., 2017, amongst others). Insights of soil water potential seasonality (posteriors in Figure 5) are critical for resolving plant hydraulic process responses to atmospheric water demand and soil water supply (Novick et al., 2019, Konings et al., 2017, Liu et al., 2021). Quantitative top-down insights into the infiltration, retention, and runoff parametrizations (histograms in Figure 3 and posteriors in Figure 5) are key for understanding the partitioning of precipitation — and its associated seasonal and inter-annual variability — into runoff and storage (which all remain key uncertainties in hydrological models). Ultimately, mechanistic insights allow for further investigations into instantaneous and lagged responses of soil hydrological states to climatic variability. Of course, these process dynamics can vary between watersheds, and it is important to understand the causes and drivers of variability in water storage between basins.

### 3.2.3 Model parameters, TWS, and states – other basins

To assure the results from the parameter inference can provide insights for other basins, we estimate parameter posteriors and corresponding TWS simulations for the two other basins mentioned above, Basins 1 (a basin that is more wet than Gavião and is located upstream from the Acanaui river gauge station) and Basin 2 (a basin that is drier than Gavião and is upstream from

the Guayaramerin river gauge station). Table S1 reports the median value for the posterior distributions of each parameter in each basin. The TWS simulations for each basin are shown in Figure S3 (Basin 1) and S4 (Basin 2). Applying the parameter inference for these basins also produced accurate simulations, with an $r^2$=0.9548 and RMSE=28.49 mm between observed and simulated TWS for Basin 1 (Figure S3), and an $r^2$=0.9891 and RMSE=18.89 mm between observed and simulated TWS for Basin 2 (Figure S4). Furthermore, we show in Figures S5 and S6 the GRACE-informed model simulated states and fluxes for Basins 1 and 2, respectively. From these results, it is apparent that Basin 1 is more wet than Basin 2, e.g., this can be seen by comparing the precipitation levels depicted in Figures S3 and S4, but also by comparing the matric potential values in Panel A or the discharge values in Panels D-E in Figures S5-S6. The location of these basins in the context of the broader South America are shown in the bottom right panel of Figures S5 and S6. Overall, the modelled state variables and parameters for these basins are constrained using the GRACE data, and this information can be used to identify and estimate the processes responsible for TWS variability in these watersheds.

### 3.3 Model simulations at the Gavião watershed: model validation, annual cycle, and annual variability

#### 3.3.1 Model calibration and validation

It is typical in works involving parameter inference to apply a model calibration and a model validation to different periods of the data set to ensure that the estimated parameters are not over-fitting the data and can be used to describe the underlying system and thus make predictions. In this section, we apply a model calibration in the Gavião watershed for the first half of the data set spanning 5 years, and then we apply a validation for the second half of the data set spanning the remaining 5 years. Figure 6 shows results for the model calibration and validation. Posterior ranges of the model simulated TWS are shown in Figure 6 in the orange envelopes for the calibration and validation years, and the red line represents the mean estimates for the validation period. The results in Figure 6 show that the calibration period RMSE is 47.71 mm with a correlation of 0.9520, and for the validation period the RMSE is 40.17 mm with a correlation of 0.9801. This shows that the estimated parameters during the calibration period are still valid for the validation period and indicates that the GRACE-informed soil hydrologic model parameters are both useful for diagnosing present-day soil water dynamics (calibration) as well as predicting seasonal and inter-annual soil water dynamics (validation).

#### 3.3.2 Annual cycle and annual variability

We further investigate the ability of the tuned model to capture the annual variability in TWS in the Gavião watershed. We compare in Figure 7A the annual cycle of the TWS anomalies produced from GRACE with those produced by the model. The annual variability is captured well with the model, with an $r^2$=0.9979 and RMSE=11.00 mm between observed and simulated TWS annual cycles. The annual cycle of the mean prior simulation is also shown in Figure 7 (red dashed line) for comparison. In Figure 7B, the timeline of de-seasonalized TWS anomaly estimates are shown. To obtain this plot, we subtract the annual cycle in Figure 7A from each month's estimate shown in Figure 4. The de-seasonalized plot in Figure 7B has an $r^2$=0.8512 and RMSE=29.27 mm between observed and simulated timelines, and the model accurately portrays whether a dry or wet period is experienced relative to what is expected in the annual cycle. This is a vast improvement from estimating the annual

cycle and de-seasonalized TWS timeline in the prior simulations (mean prior simulation shown in Figure 7, and the distribution of prior simulations is shown in Figure S7). For the prior simulations of the annual cycle, the model has a $r^2$=0.9761 and RMSE=417.62 mm, and for the de-seasonalized TWS timeline, the model prior has a $r^2$=0.4323 and RMSE=93.32 mm between observed and simulated timelines. Therefore, the model posterior solutions show a great improvement from the prior for simulating the annual cycle and capturing the seasonality of the hydrologic cycle for each watershed.

In the results shown in Figure 7, we see that the model can capture the 2005-2006 and 2010-2011 droughts in the Gavião watershed that are shown in the GRACE data (c.f. Lewis et al., 2011). The model also captures the wet periods observed in 2003, 2004, 2008, 2009, and 2012 (see Figure 7B). The model captures the positive and negative anomalies quite well; however, it does have some limitation in capturing the magnitude of some extreme events (positive and negative), which may be partly caused by the coarser time step and spatial scale of the simulation. Yet, the model does succeed in capturing some

delayed anomalies in water storage following the 2005-2006 and 2010-2011 droughts, which is very promising. This gives confidence in the data-constrained model to provides meaningful estimates of TWS anomalies on monthly and seasonal scales.

### 3.4 Correlations between posterior model parameters and model states

After the model parameters and states variables are constrained by the GRACE data for the Gavião watershed, relationships between the model parameters and simulated states begin to emerge. We show in Figure 8A the scatter plot between posterior

solutions of model simulated TWS and the excess runoff parameter. This figure shows that the region inside the black box, or the high-density region of the posterior, is the region within the posterior domain that has high information content (i.e., plausible solutions with high likelihood). The true value provided by the GRACE data is marked with a red line in Figure 8A. Other regions of this space, such as locations with excess runoff values below 0.2, produce unlikely model simulations, and similarly locations with excess runoff values higher than 0.5 are also less likely. This can also be seen in Figure 3, in the

posterior histograms for the excess runoff parameter. Similar relationships between other parameters and state variables (including soil moisture of layer 1 and discharge from layer 1) are shown in Figure S8 of the supplementary section. Overall, these plots not only show the emergent relationships between variables as informed by GRACE, but also indicate if and how they are correlated in the Gavião watershed.

Similarly, the posterior parameter solutions can be used to infer relationships between the parameters themselves. To

400 this end, we show in Figure 8B a scatter plot depicting the GRACE-informed correlation of the posterior parameter values for the soil moisture initialization parameters in the Gavião watershed. We see that the initial soil moisture in the bottom layer is greater than 0.2 [$m^3/m^3$], and in the top layer is less than 0.3 [$m^3/m^3$], which can be seen in Figure 3, in the posterior histograms for the soil moisture initialization parameters. One property that also emerges in Figure 8B is that the initial soil moisture in the bottom layer is larger than that of the top layer. This indicates that, in the initial time step of the simulations, the bottom

layer should have a higher soil moisture than the top layer. These relationships can be created for any pair of parameters in the posterior space, and Figure S9 of the supplementary section portrays these relationships for several combinations of parameters, indicating what combinations of parameters are possible for this hydrologic system, as inferred by GRACE.

We summarize the results reported in this subsection with the following points. First, we find considerable correlations between the posteriors of individual model parameters and model states in the Gavião watershed. We also find considerable correlations between the posteriors of individual model parameters and with other parameters. This is important, because the correlations between parameters and states indicate that the choice of hydrological constants can have a considerable impact on simulated TWS. The relationships found in the parameter posteriors imply that while several parameters exhibit considerable uncertainty, only a subset of parameter combinations provide GRACE-consistent model solutions. In essence, these GRACE-based relationships portray what parameter combinations are possible for accurately simulating the chosen watershed.

## 4 Summary

In this paper we used a parsimonious hydrologic model capable of simulating various aspects of land surface hydrology, and we ran the model for different basins in the western Amazon. We performed extensive analysis on the Gavião watershed, a relatively wet basin, and also reported results for two other basins, one being more wet (Basin 1) and one being more dry (Basin 2). The model used in this study includes two soil layers (plant available and unavailable water pools), is driven with hydrologic flux variables such as evapotranspiration and precipitation and includes other processes such as infiltration into the soil, surface runoff, drainage from each layer, and recharge into the lower soil layer. Listed in Table 1 are various model parameters that control the simulations for the Gavião watershed, with their respective estimated values. Table S1 lists these parameter values for Basins 1 and 2. We applied Bayesian inference to estimate posteriors for the model parameters that allowed the simulations to match satellite-based estimates of Total Water Storage (TWS) obtained from GRACE.

Results in this paper showcased the estimated parameter posteriors along with their priors (Figure 3), the posterior solution of simulated TWS (Figure 4), and the estimated model states (Figure 5). We also performed a model calibration and validation exercise (Figure 6), to show how estimated parameters during the calibration period are still useful for the validation period. We also compared the annual cycle and de-seasonalized TWS anomalies produced from both the GRACE data and the model, and we showed how the data constrained TWS model can capture the annual variability as well as drought events that occurred in this system (Figure 7AB). For further diagnosis of our results, we showed the relationships between model simulated states and the estimated parameters (Figure 8A and Figure S8). Then we showed relationships between combinations of estimated parameters (Figure 8B and Figure S9). Furthermore, we investigated the sensitivity of the model simulated TWS to minor perturbations in parameter values (Figure 2 and Figure S1), and we showed how parameters can create sensitivities in TWS in different ways, for example during wet or dry periods, or during model initialization. Simulation results for Basins 1 and 2 are shown in supplementary Figures S3-S6.

Overall, the results in this paper allowed us to make the following conclusions. First, GRACE-informed soil hydrologic model parameters are useful for diagnosing present-day soil water hydrology. Substantial uncertainty reduction was found for parameters that represent soil moisture initialization, rooting depth, and conductivity and retention relationships. However, limited uncertainty reduction was found for infiltration rates and porosity parameters, and further model

development may be needed to describe the information content of these processes and their associated uncertainties more accurately. The second conclusion is that GRACE-informed model parameters can be used for predicting seasonal and inter-annual soil water hydrology. We showed that using a 5-year data record of TWS allows the parameter inference to still be applicable to the remaining 5-year data record, which is simulated without the use of information from GRACE. Lastly, a medium complexity model like the one used here can be sufficient for capturing monthly to seasonal-scale hydrology of the land surface at the basin scale, such as the Gavião watershed in the Amazon.

By fusing information from the signal of the surface mass change with other hydrologic information, such as physical constrains in model simulations or seasonal behaviour of in-situ observations, GRACE has proven its ability to infer hydrologic parameters and state variables accurately. We found that this methodology is generalizable to other regions, and we reported the results from additional testing that was conducted to other watersheds in the Amazon. Our results suggest the potential of using gravimetric observations of TWS from GRACE to identify and constrain key parameters in soil hydrologic models.

*Code availability*. The codes are available by contacting the authors.

*Data availability*. All data used in this work are of public domain and freely downloadable at the links quoted in the text.

*Author contributions*. EM led the investigation, conceptualized the research, did the formal analysis and model simulations, and wrote the original draft. AB took responsibility for the investigation, developed the methodology, conceptualized the research, acquired the funding and the resources, and reviewed and edited the paper. ML developed the methodology, conceptualized the research, and reviewed and edited the paper. JR and PL reviewed and edited the paper. JW did the formal analysis, took responsibility for the investigation, reviewed and edited the paper, and supervised the project.

*Competing interests*. The authors declare that they have no conflict of interest.

*Acknowledgments*. This research was carried out at the Jet Propulsion Laboratory, California Institute of Technology, under a contract with the National Aeronautics and Space Administration (80NM0018D0004). Copyright 2021. A portion of this work was supported by funding from the NASA GRACE-FO Science team. M.L. was supported by the NASA Postdoctoral Program, administered by Universities Space Research Association under contract with NASA. The authors thank A. Konings for insightful discussions that helped form the concepts presented in this paper. GRACE data is available at https://grace.jpl.nasa.gov/data/get-data/.

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

| Parameter | Symbol | Min | Max | Units | MCMC | AK Diagonal | TWS Sensitivity |
|---|---|---|---|---|---|---|---|
| 1) Porosity Layer 1 | $\rho_1$ | 0.2 | 0.8 | | 0.4686 | 0.0509 | 0.1192 |
| 2) Porosity Layer 2 | $\rho_2$ | 0.2 | 0.8 | | 0.4544 | 0.0127 | 0.0614 |
| 3) Ψ_field | $\Psi_{field}$ | -0.1 | -0.01 | MPa | -0.0375 | 0.3735 | 0.3656 |
| 4) Layer 1 Depth (Rooting Depth) | $L_{PAW}$ | 1 | 100 | m | 23.7214 | 0.8441 | 0.2262 |
| 5) Layer 2 Depth (PUW Depth) | $L_{PUW}$ | 1 | 100 | m | 12.2266 | 0.7136 | 0.5975 |
| 6) Retention Parameter b | b | 1.5 | 10 | | 2.3767 | 0.8448 | 0.3647 |
| 7) Saturated Hydraulic Conductivity | $K_0$ | 1.00E-07 | 1.00E-05 | m/s | 2.57E-06 | 0.2593 | 0.4455 |
| 8) Maximum Infiltration | $I_{max}$ | 100 | 2000 | mm/month | 1275.9 | 0.7758 | 0.0485 |
| 9) SM@t=0 PAW | $SM_{1,t0}$ | 0.1 | 0.5 | $m^3/m^3$ | 0.1607 | 0.5873 | 0.3128 |
| 10) SM@t=0 PUW | $SM_{2,t0}$ | 0.1 | 0.5 | $m^3/m^3$ | 0.4117 | 0.7889 | 0.133 |
| 11) ET scale factor | $ET_{scale}$ | 0.5 | 1.5 | | 0.5364 | 0.9694 | 0.0179 |
| 12) P scale factor | $P_{scale}$ | 0.5 | 1.5 | | 0.8284 | 0.897 | 0.0586 |
| 13) Q excess factor | $Q_{excess}$ | 0.01 | 1 | | 0.2832 | 0.864 | 0.0246 |

**Table 1:** Parameter estimation results for the Gavião watershed. Shown here are the model parameters and associated symbols, prior ranges (Min – Max), units, Posterior solution median estimate (MCMC), AK matrix diagonal values showing the level of uncertainty reduction (i.e., AK=1 for full reduction, AK=0 for no reduction in uncertainty), and TWS sensitivities ([mm change in TWS per 1%-unit change in parameter]) showing the sensitivity of TWS variability to model parameters.


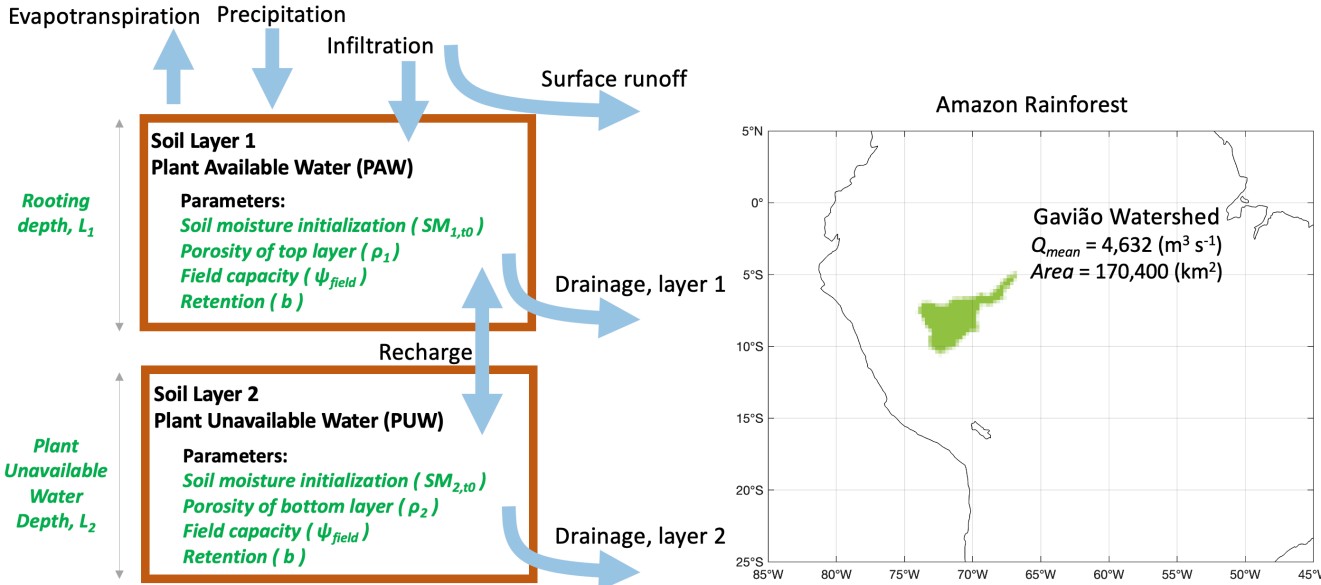

**Figure 1:** Model schematic for the data-constrained terrestrial water storage model. Arrows indicate the logical flow that describes the movement and storage of water in the model. The domain on the right highlights the western Amazonian watershed investigated in this study, the Gavião Watershed.

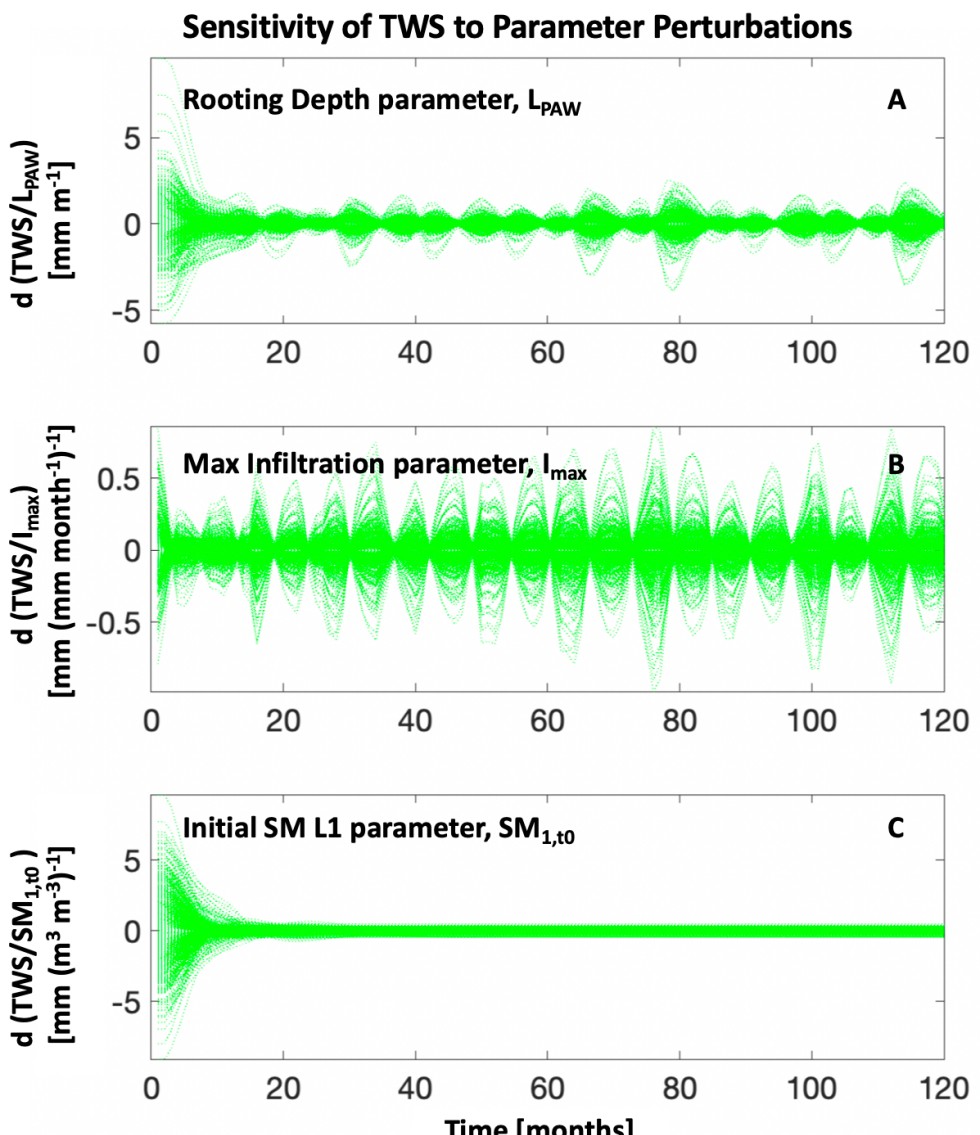

**Figure 2:** Sensitivity of the model simulated TWS to minor perturbations in parameter values. Shown here (from top to bottom) are sensitivities to A) the Rooting depth parameter, B) the Maximum infiltration parameter, and C) the soil moisture initialization parameter for layer 1. Green curves are the changes in simulated TWS (d TWS) when each parameter is perturbed (d Par) by 1% of its prior range, indicating the magnitude and the time steps of model sensitivity. TWS sensitivities to other parameters are shown in Figure S3 of the supplementary section. The x-axis depicts the number of months since 2003, showing the ten-year period starting in January 2003 and ending in December 2012.

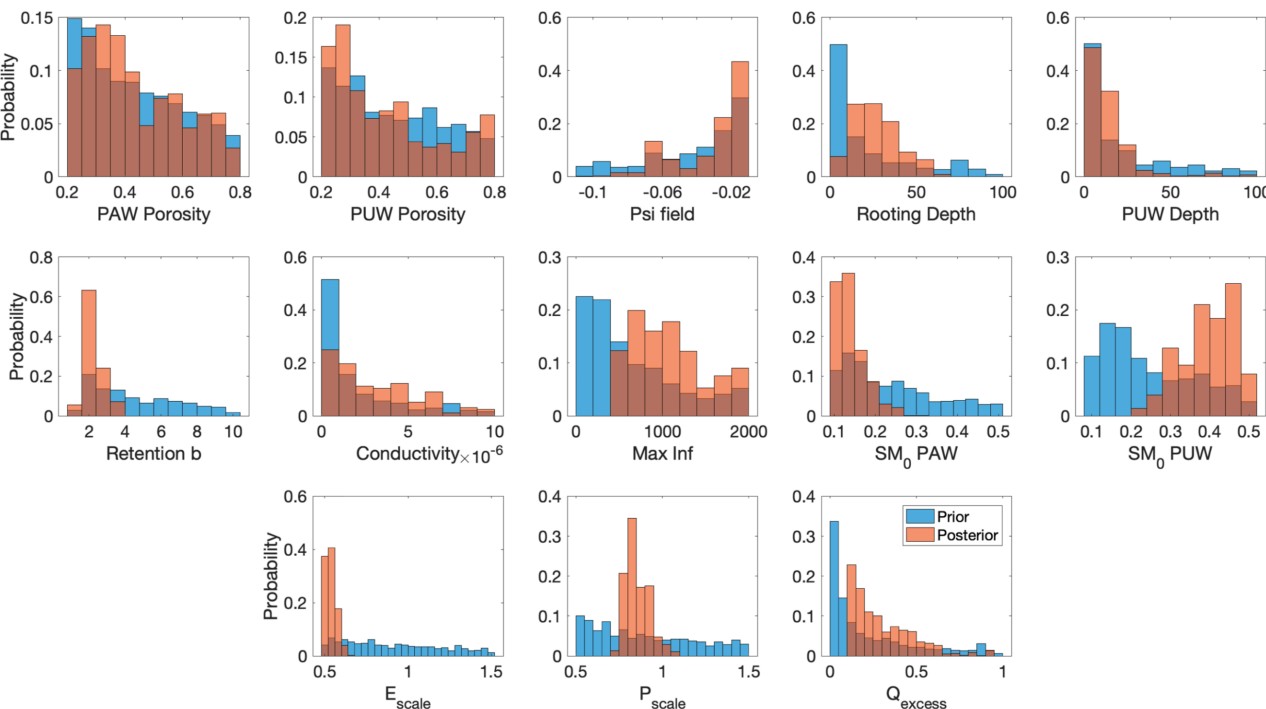

**Figure 3:** Histograms of the prior (blue) and posterior (orange) distributions of the GRACE-informed parameters for the Gavião watershed.

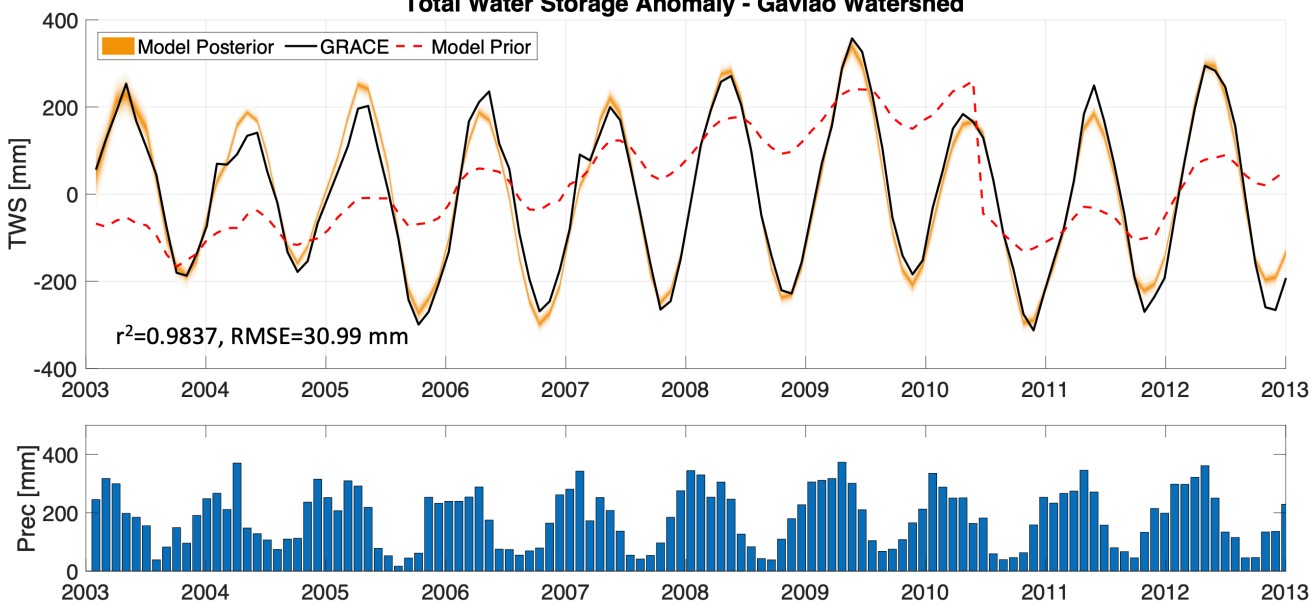

**Figure 4:** Monthly Total Water Storage (TWS) anomaly estimates from satellite data (GRACE TWS), the prior simulation from the model, and the data-constrained version of the model simulations for the Gavião watershed. GRACE-informed posterior ranges of the model simulated TWS are shown here in the orange envelopes. Precipitation values used to drive the model are shown to indicate the seasonal cycle.

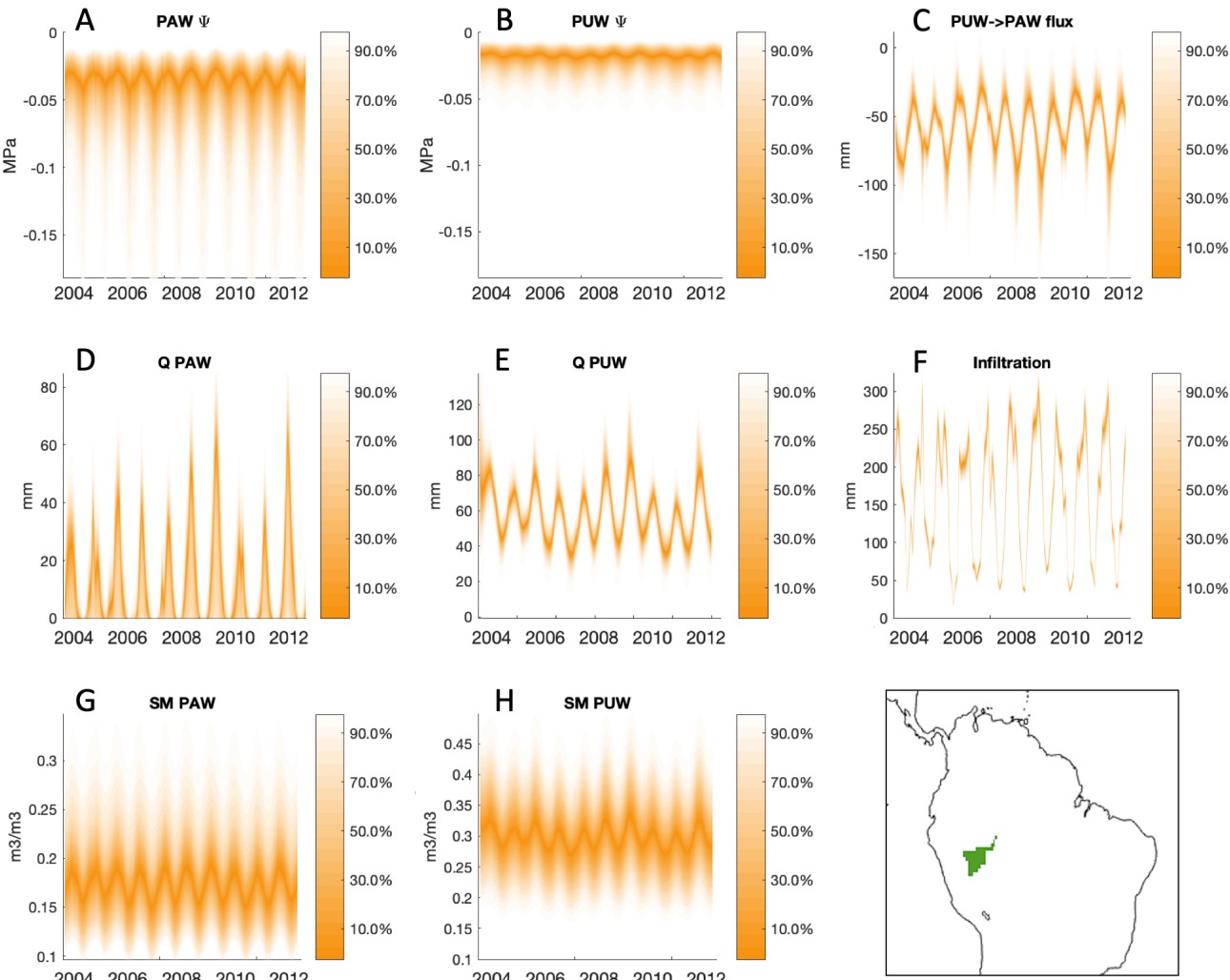

**Figure 5:** GRACE-informed model simulated states and fluxes for the Gavião watershed (basin shown in the bottom right panel in the context of the broader South American domain). These figures show specific model processes, such as (A) the matric potential of plant available water (PAW ψ), (B) the matric potential of plant unavailable water (PUW ψ), (C) recharge (PUW -> PAW flux) where negative values indicate a downwards flux, (D) discharge from the top layer (Q PAW), (E) discharge from the bottom layer (Q PUW), (F) infiltration, (G) soil moisture of the top layer (SM PAW), and (H) soil moisture of the bottom layer (SM PUW). The ranges shown here in orange envelopes indicate the GRACE-informed posterior ranges. A map showing the location of the Gavião watershed is shown in the bottom right panel.

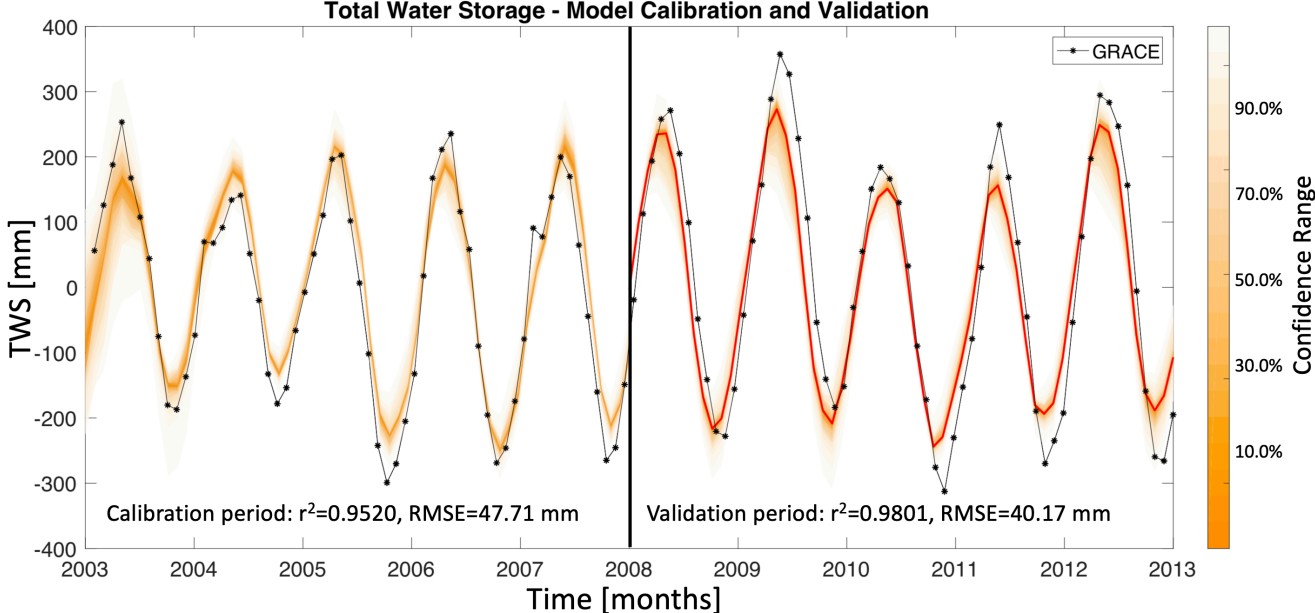

**Figure 6:** Model calibration and validation for monthly TWS anomaly estimates in the Gavião watershed, for the period
January 2003 through December 2012. The plot shows the first 5 years of the data for calibration and the remaining 5 years
for validation. GRACE-informed posterior ranges of the model simulated TWS are shown here in the orange envelopes for
the calibration and validation years, and the red line is used to represent the mean estimates for the validation period.

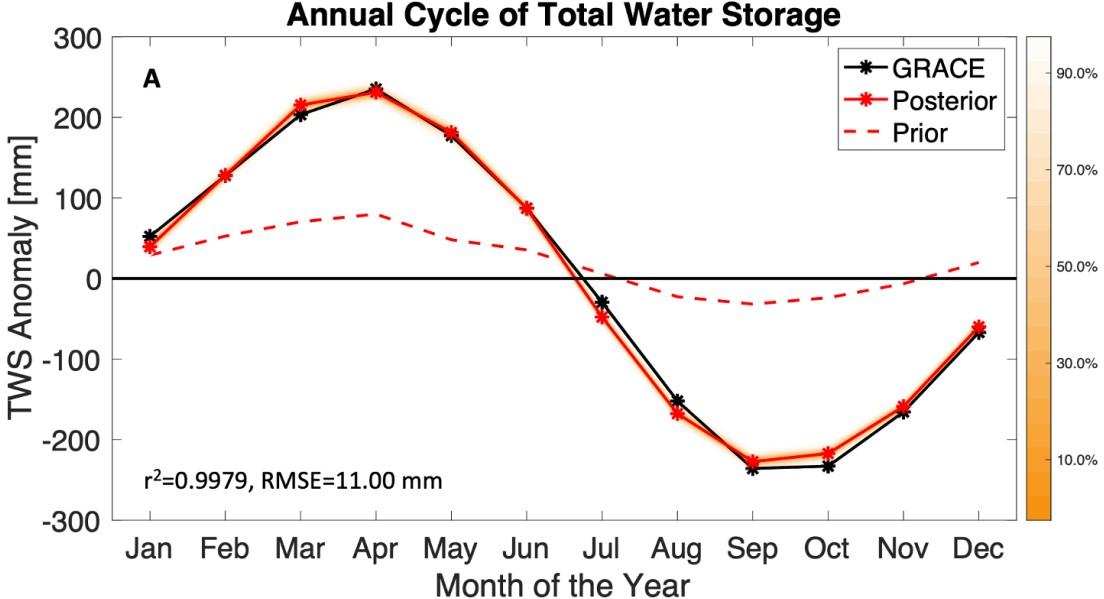

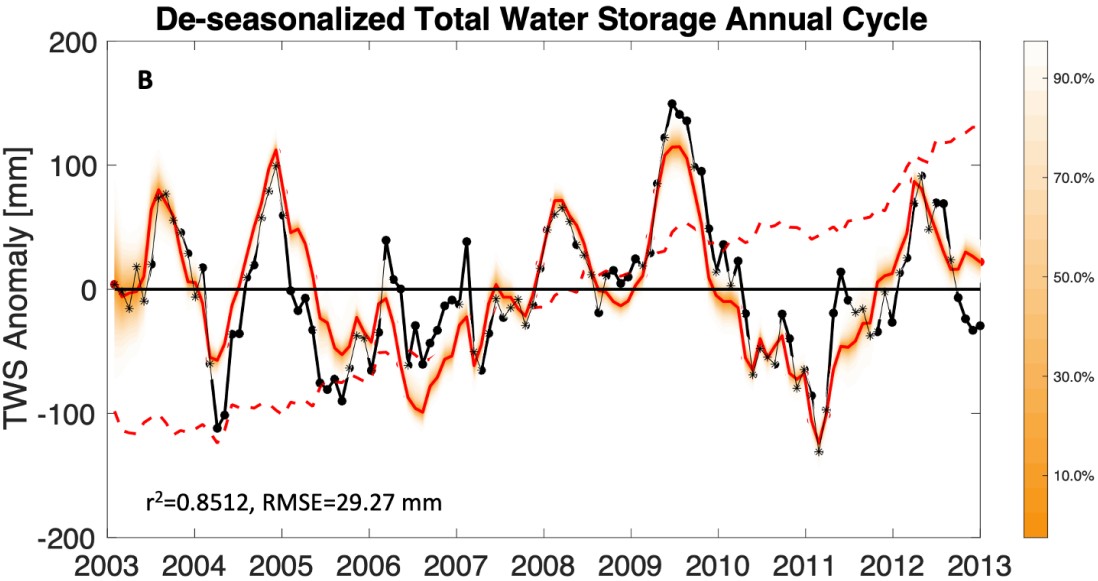

**Figure 7:** A) Annual cycle of the monthly TWS anomalies [mm], from satellite data (GRACE), the prior simulation from the
model (Prior), and the data-constrained version of the model simulations (Posterior) for the Gavião watershed. GRACE-
informed posterior ranges of the model simulated TWS annual cycle are shown here in the orange envelopes. B) To obtain
the de-seasonalized values of TWS for the Gavião watershed shown in Panel B, we subtract the annual cycle in Panel A from
each month's estimate shown in Figure 4. This shows whether the anomaly values in each time step of Panel B portrays an
extremely dry or wet period relative to what is expected in the annual cycle. Hence, the data-constrained model can capture
the 2005-2006 and 2010-2011 droughts that are shown in the GRACE data.

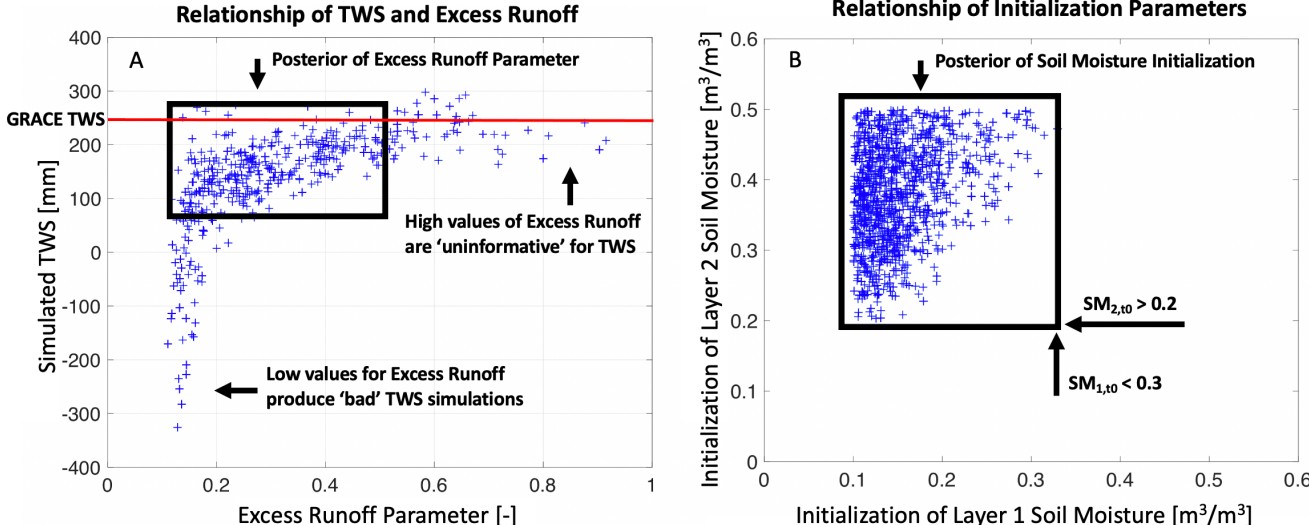

**Figure 8:** A) Posterior relationship of the model simulated TWS [mm] during April 2003 and the runoff excess parameter [unitless]. The region inside the black box indicates the posterior region with high density, i.e., plausible solutions with high likelihood. The red line shows the 'true' TWS value seen in the GRACE data for this period. B) Posterior relationship of the initialization parameters for soil moisture in layers 1 and 2, respectively. Initial SM in layer 2 is larger than 0.2 [m³/m³], initial SM in layer 1 is less than 0.3 [m³/m³], and $SM_{2,t0}$ is generally larger than $SM_{1,t0}$, as indicated in this plot. See Table 1 and Section 3.4 for details.