# Peer review of "Information content of soil hydrology in a West Amazon watershed as informed by GRACE"

_Hydrology and Earth System Sciences, 2021_

## Author Comment (AC1)

Author response to:
RC1: 'Comment on hess-2021-104', Anonymous Referee #1, 11 Apr 2021

This study applied the Markov Chain Monte Carolo algorithm to calibrate a simple water balance model using GRACE TWS observations. The posterior model parameters, model states and simulated TWS for one watershed were shown in this study. The results suggested the potential of using GRACE to constrain model parameters. The topic is relevant for reader of HESS. However, I believe some critical points need to be clarified and supported by additional results.

Thank you for your time spent reviewing our manuscript and for your valuable comments. We hope our notes below address your concerns about this paper.

1. It was mentioned that the ET in the model was derived from the satellite observations of precipitation and TWS and ground-based river runoff (L138). I assumed that it must be GRACE TWS here. If so, GRACE data has been used in the modelling and resued in the model calibration through MCMC. The GRACE data was not independent to the model simulations. Please clarify.

This is true. To clarify this more, there are 3 different derivations used for the TWS variable. These 3 estimates provide a sense of uncertainty for the TWS. The uncertainty from these different 'products' is used in the likelihood function of the MCMC algorithm when fitting the model simulated TWS to the GRACE derived TWS. Then, there is also 3 products used in the precipitation and the runoff driving variables that were used, to get a sense of the uncertainty in each variable. To estimate the ET driving variable in this work, we use the mean of the TWS, P, and Q products and create a water balance that will allow us to estimate a mean for the ET driving variable. Then by application of the ET scaling parameter, we try to estimate whether our initial calculation of ET required any scaling to match the data. Therefore, even though the GRACE TWS is somehow used in the derivation of the ET data, the uncertainty that is applied throughout the work allows us to still estimate ET that is not dependent on the GRACE data. We will provide these details in the new version of the manuscript to provide more clarity to the readers.

2. The results for one watershed is not convincing. In particular, it was mentioned that additional testing over other watershed has been done in L369. Results for few more watersheds can help supporting this study. Since the inputs for the model are satellite rainfall, TWS and in-situ river runoff, it would be surprise there is no other watershed has enough river runoff data over Amazon.

Thank you for this insight. We will include results of the model simulated TWS as well as the inferred parameter values from two other basins in the Amazon. By examining the differences in the parameter inference and resulting TWS simulations between the different basins, we believe we can better answer the questions laid out in the paper. We will choose two other watersheds with contrasting precipitation levels, to demonstrate how information content varies across precipitation gradients.

3. Results of model simulations without MCMC should be compared with the posterior TWS to demonstrate the improved performance.

We again agree with the comment here and will include the prior simulation of TWS to show the improved performance after applying MCMC.

4. L76: Is the model proposed by the authors for the first time? Or any reference for the model?

Yes, this is the first time this model is reported in the literature. We will clarify this in the revised manuscript

5. L138: Was the satellite precipitation from TRMM here as well?

Yes, the precipitation product is from TRMM. We will clarify this in the revised manuscript

6. L173: Spherical harmonic solutions here?

Yes, spherical harmonics solutions used here. We will clarify this in the revised manuscript

7. L260: The information about each variable for each subfigure has been included in the figure caption. More discussion about the results instead would be helpful.

Yes, we can replace these comments with additional discussion about the results. Specifically, we can provide more comments about the supplementary figures which show how the parameter posteriors correlate with one another. These results in the supplement indicate what parameters combinations are ultimately possible, and more discussion about these combinations can be useful for the reader.

8. L285-290: could you include the r2 for the model simulated TWS and de-seasonalized TWS without MCMC as a comparison? Also plot the time-series together in Fig6?

Yes, we can add these to the figures. And we can combine the figures together in one. We will do this in the revised manuscript.

9. The sensitivity results from figure 8 would make more sense to mentioned in the beginning of the section since the results were summarised in Table1.

This can be moved to the beginning of the section. We will do this in the revised manuscript.

10. Changing x-axis to years Figure 4 and 8 like the other plots would be more reader-friendly.

We agree with this comment. We'll change the x-axis in all figures to represent years.

---

## Author Comment (AC2)

Author response to:
RC2: 'Comment on hess-2021-104', Anonymous Referee #2, 17 May 2021

The article is well written and all the concepts are explained in required details. However, the inference drawn from the study need more clarifications and probably a major revision. After writing this review, I happened to check the review comments from reviewer 1 and I had exactly the same comments as their comment no 1, 2, 3 and 10. I am not repeating them here. My additional concerns are:

Thank you for your time spent reviewing our manuscript and for your valuable comments. We hope our notes written to the other reviewer that address their comments 1, 2, 3, and 10 also address your concerns about this paper.

1) Authors have stated, in the title and in the introduction (line 65 to 69) that they are able to identify and estimate the processes responsible for TWS variability. Explicitly they have stated their aims as: a)"inform and reduce uncertainties of terrestrial hydrologic processes regulating the seasonal and inter-annual variability of TWS in the western Amazon, the Gavião watershed" b)Develop a model to represent the first order controls on seasonal-to-decadal soil moisture dynamics, and c) employ a Bayesian model-data fusion approach to constrain model parameters (namely initial states and time-invariant process variables), such that differences between GRACE and simulated TWS anomalies are statistically minimized

Therefore the first impression is that the study will help us understand the cause and drivers of variability in the water storage. However, the results and summary lack discussion on the parameters and physical processes that drive the dry season or wet season, and why?

This is a great comment. We believe we can address this issue with the inclusion of additional results from two other basins. By examining the differences in the parameter inference and resulting TWS simulations between the different basins, we believe we can better answer the questions laid out in the introduction. We will choose two other watersheds with contrasting precipitation levels, to demonstrate how information content varies across precipitation gradients.

2) The calculation of basin average TWS from GRACE data is not clearly explained. Did the authors use mascon data or spherical harmonic data? How do they combine these three solutions? How do they tackle the limitations of the coarse spatial resolution of GRACE data and the different spatial resolution of the three data products.

Thank you again for this comment. Yes, we use the spherical harmonic data. We aggregate the GRACE information over the entire basin and estimate a basin-wide timeline of the TWS anomalies. We apply the same method to aggregate the other products to estimate a basin-wide timeline of P and Q, and therefore estimate a basin-wide timeline for the ET variable as well.

3) Can authors comment or compute their model for an arid or semi-arid region or a river catchment with trends (such as the Great Basin in the USA)? What do they expect to obtain from such a model at global scale?

Yes, we will apply a similar computation for simulations in two other basins in the Amazon. We will choose two other watersheds with contrasting precipitation levels, to demonstrate how information content varies across precipitation gradients.

4) Please mention the catchment name in the title instead of 'Amazon' or compute the model for whole Amazon.

As mentioned, we will apply the methodology for 2 other basins in the amazon. We will change the title to reflect this, i.e. "Information content of soil hydrology in Amazonian watersheds as informed by GRACE".

5) Line 361: Authors state "GRACE-informed model parameters can be used for predicting seasonal and inter-annual soil water hydrology in the absence of concurrent GRACE measurements. We showed that using a 5-year data record of TWS allows the parameter inference to still be applicable to the remaining 5-year data record, which is simulated without the use of information from GRACE". If the evapotranspiration product was obtained using GRACE, then the simulated part is not completely free from GRACE.

We addressed a similar comment by reviewer 1, and we will re-address this issue here. To clarify this more, there are 3 different derivations used for the TWS variable. These 3 estimates provide a sense of uncertainty for the TWS. The uncertainty from these different 'products' is used in the likelihood function of the MCMC algorithm when fitting the model simulated TWS to the GRACE derived TWS. Then, there is also 3 products used in the precipitation and the runoff driving variables that were used, to get a sense of the uncertainty in each variable. To estimate the ET driving variable in this work, we use the mean of the TWS, P, and Q products and create a water balance that will allow us to estimate a mean for the ET driving variable. Then by application of the ET scaling parameter, we try to estimate whether our initial calculation of ET required any scaling to match the data. Therefore, even though the GRACE TWS is somehow used in the derivation of the ET data, the uncertainty that is applied throughout the work allows us to still estimate ET that is not dependent on the GRACE data. To address this specific comment of the reviewer, though, we will change the terminology in this sentence, and will not state that "*GRACE-informed model parameters can be used for predicting seasonal and inter-annual soil water hydrology in the absence of concurrent GRACE measurements*". Instead, we will simply refer to this part of the manuscript as a calibration/validation exercise.

---

## Author Response (AR1)

**Author response to:**
**RC1: 'Comment on hess-2021-104', Anonymous Referee #1, 11 Apr 2021**

This study applied the Markov Chain Monte Carolo algorithm to calibrate a simple water balance model using GRACE TWS observations. The posterior model parameters, model states and simulated TWS for one watershed were shown in this study. The results suggested the potential of using GRACE to constrain model parameters. The topic is relevant for reader of HESS. However, I believe some critical points need to be clarified and supported by additional results.

Thank you for your time spent reviewing our manuscript and for your valuable comments. We hope our notes below address your concerns about this paper.

1. It was mentioned that the ET in the model was derived from the satellite observations of precipitation and TWS and ground-based river runoff (L138). I assumed that it must be GRACE TWS here. If so, GRACE data has been used in the modelling and resued in the model calibration through MCMC. The GRACE data was not independent to the model simulations. Please clarify.

This is indeed the case. We now explicitly clarify that the ET estimates are based on satellite-based estimates of P, TWS and Q in Lines 147-159. To further clarify our approach, we indicate that there are 3 different derivations used for the TWS variable. While we agree with the reviewer that GRACE is not fully independent of the model simulations, we clarify that the satellite-based ET is optimized—along with other parameters and initial conditions—to minimize the model-data mismatch. Finally, we now also add a sentence on Lines 321-324 of the main text to highlight alternative approaches for prescribing watershed ET in future efforts.

2. The results for one watershed is not convincing. In particular, it was mentioned that additional testing over other watershed has been done in L369. Results for few more watersheds can help supporting this study. Since the inputs for the model are satellite rainfall, TWS and in-situ river runoff, it would be surprise there is no other watershed has enough river runoff data over Amazon.

Thank you for this insight. We have now included results of the model simulated TWS as well as the inferred parameter values from two other basins in the Amazon. By examining the differences in the parameter inference and resulting TWS simulations between the different basins, we provided some answers the questions laid out in the paper. We chose two other watersheds with contrasting precipitation levels, to demonstrate how information content varies across precipitation gradients. These watersheds are located upstream from the Acanaui river gauge station (hereafter called Basin 1) and upstream from the Guayaramerin river gauge station (hereafter called Basin 2). Results for these watersheds are shown in Figures S3-S6 and discussed in Section 3.2.3 on Lines 337-350.

3. Results of model simulations without MCMC should be compared with the posterior TWS to demonstrate the improved performance.

We agree with the comment here and include the prior simulation of TWS to show the improved performance after applying MCMC. The prior solutions are shown in Figures S2. These results are discussed in the manuscript on Lines 294-298. In essence, by applying the MCMC scheme without using the GRACE TWS data, we sampled from the prior parameter and initial conditions, and found that prior model simulations show little—if any skill—when compared against GRACE data. Therefore, comparing the result of the posterior model simulations with the prior model simulations, we see a major improvement in the constrained posterior model simulations.

4. L76: Is the model proposed by the authors for the first time? Or any reference for the model?

We now indicate on Lines 158-159 that is the first time this model is reported in the literature.

5. L138: Was the satellite precipitation from TRMM here as well?

Yes, the precipitation product is from TRMM. We clarify this in the paper on Lines 137-139.

6. L173: Spherical harmonic solutions here?

Yes, spherical harmonics solutions used here. We clarify this in the paper on Line 184.

7. L260: The information about each variable for each subfigure has been included in the figure caption. More discussion about the results instead would be helpful.

Yes, we have replaced these comments with additional discussion about the results. These comments can be seen in the revised manuscript on Lines 299-320.

8. L285-290: could you include the r2 for the model simulated TWS and de-seasonalized TWS without MCMC as a comparison? Also plot the time-series together in Fig6?

We have added these. In the revised manuscript, we show the prior simulations of the annual cycle and de-seasonalized TWS in Figure S7 and we report the r2 and RMSE for these timelines on Lines 372-374.

9. The sensitivity results from figure 8 would make more sense to mentioned in the beginning of the section since the results were summarised in Table1.

This has been moved to the beginning of the section. In the revised manuscript, this figure is now Figure 2 and it is discussed in Section 3.1 (Lines 258-269).

10. Changing x-axis to years Figure 4 and 8 like the other plots would be more reader-friendly.

We agree with this comment. We changed the x-axis in all figures to represent years.

**Author response to:**
**RC2: 'Comment on hess-2021-104', Anonymous Referee #2, 17 May 2021**

The article is well written and all the concepts are explained in required details. However, the inference drawn from the study need more clarifications and probably a major revision. After writing this review, I happened to check the review comments from reviewer 1 and I had exactly the same comments as their comment no 1, 2, 3 and 10. I am not repeating them here. My additional concerns are:

Thank you for your time spent reviewing our manuscript and for your valuable comments. We hope our notes written to the other reviewer that address their comments 1, 2, 3, and 10 also address your concerns about this paper.

1) Authors have stated, in the title and in the introduction (line 65 to 69) that they are able to identify and estimate the processes responsible for TWS variability. Explicitly they have stated their aims as: a)"inform and reduce uncertainties of terrestrial hydrologic processes regulating the seasonal and inter-annual variability of TWS in the western Amazon, the Gavião watershed" b)Develop a model to represent the first order controls on seasonal-to-decadal soil moisture dynamics, and c) employ a Bayesian model-data fusion approach to constrain model parameters (namely initial states and time-invariant process variables), such that differences between GRACE and simulated TWS anomalies are statistically minimized

Therefore the first impression is that the study will help us understand the cause and drivers of variability in the water storage. However, the results and summary lack discussion on the parameters and physical processes that drive the dry season or wet season, and why?

We agree with the reviewer that an overview of key insights was lacking in the previous version of the paper. We have now added additional analysis and text in the paper to address this issue. We have added a new section on Lines 325-336 (Section 3.2.2 Interpretation of results) to highlight some of the quantitative insights on the role of soil states and processes in regulating seasonal-to-decadal soil moisture supply. We have also added two additional watershed analyses on Lines 337-350 (Section 3.2.3 Model parameters, TWS, and states – other basins) and in the supplementary section, where we applied the methodology to Basin 1 (a more wet basin located upstream from the Acanaui river gauge station) and Basin 2 (a drier basin located upstream from the Guayaramerin river gauge station). With the addition of these results, we investigate and discuss the role of process variability across these watersheds.

2) The calculation of basin average TWS from GRACE data is not clearly explained. Did the authors use mascon data or spherical harmonic data? How do they combine these three solutions? How do they tackle the limitations of the coarse spatial resolution of GRACE data and the different spatial resolution of the three data products.

Thank you for this comment. Yes, we use the spherical harmonic data, and this is mentioned in the text on Line 184. We aggregate the GRACE information over the entire basin and estimate a

basin-wide timeline of the TWS anomalies. We apply the same method to aggregate the other products to estimate a basin-wide timeline of P and Q, and therefore estimate a basin-wide timeline for the ET variable.

3) Can authors comment or compute their model for an arid or semi-arid region or a river catchment with trends (such as the Great Basin in the USA)? What do they expect to obtain from such a model at global scale?

Yes, we have applied a similar computation for simulations in two other basins in the Amazon. We chose two other watersheds with contrasting precipitation levels, to demonstrate how information content varies across precipitation gradients. Results for these watersheds are shown in Figures S3-S6 and discussed in Section 3.2.3 on Lines 337-350. In principle, the model can be applied to the US or other domains, but this would require the addition of other processes not currently represented in the model, such as snow and irrigation management, which may be key methodological steps to accurately close the water balance in other regions.

4) Please mention the catchment name in the title instead of 'Amazon' or compute the model for whole Amazon.

Thank you for this comment. We have changed the title to reflect this, i.e. "Information content of soil hydrology in a West Amazon watershed as informed by GRACE".

5) Line 361: Authors state "GRACE-informed model parameters can be used for predicting seasonal and inter-annual soil water hydrology in the absence of concurrent GRACE measurements. We showed that using a 5-year data record of TWS allows the parameter inference to still be applicable to the remaining 5-year data record, which is simulated without the use of information from GRACE". If the evapotranspiration product was obtained using GRACE, then the simulated part is not completely free from GRACE.

We agree with the reviewer and have addressed a similar comment by reviewer 1. We will re-address this issue here. We now explicitly clarify that the ET estimates are based on satellite-based estimates of P, TWS and Q in Lines 147-159. To further clarify our approach, we indicate that there are 3 different derivations used for the TWS variable. While we agree with the reviewer that GRACE is not fully independent of the model simulations, we clarify that the satellite-based ET is optimized—along with other parameters and initial conditions—to minimize the model-data mismatch. Finally, we now also add a sentence on Lines 321-324 of the main text to highlight alternative approaches for prescribing watershed ET in future efforts. To address the reviewer comment that "the simulated part is not completely free from GRACE", we have changed the terminology in the text, and removed the statement that "*GRACE-informed model parameters can be used for predicting seasonal and inter-annual soil water hydrology in the absence of concurrent GRACE measurements*". Instead, we simply refer to this part of the manuscript as a calibration/validation exercise.

---

## Author Response (AR2)

**Author response to:**
**RC1: 'Comment on hess-2021-104', Anonymous Referee #1, Dec 14 2021**

The authors have addressed most of my previous comments. Some minor suggestions below for the authors to consider:

Thank you for your time spent reviewing our manuscript and for your valuable comments. We hope our notes below address your concerns about this paper.

1. L54: water instead of H2O?
We have fixed this. Thank you for the comment.

2. Theta was mentioned here as model weights in L207 but as parameters in L222. Please clarify
We have fixed this by changing 'weights' to 'parameters' to maintain consistency in the description. Thank you for the comment.

3. L226: unbold 'and'
We have fixed this. Thank you for the comment.

4. equation 13 and 14: mean of the TWS for all times not average for month t?
We have fixed this by removing the 't' subscript in the last terms of Eq 13 and 14. Thank you for the comment.

5. L295: The comparison with prior simulations is one of the main results in this study. I suggest include the subfigure A of Figure S2 in Figure 4. It can be plotted together in the top subfigure showing only the mean of the prior. Include the subfigure labels.
Thank you for the comment. We now include the prior simulations in Figure 4.

6. Figure 5: maybe no need to show the catchment location figure since it has been shown in Figure1.
Although we agree with the reviewer that this map is repeated here, we prefer to keep the subpanel to match the maps shown in supplementary Figures S5-S6.

7. Figure 7: same suggestion here. it would be good to include the prior simulations in Figure S7 here as a comparison. Authors refer to many figures in supplementary materials which can be included in the main manuscript.
Thank you for the comment. We now include the prior simulations in Figure 7.

8. Table 1: no need to have bold font for all the parameters and values.
We have fixed this. Thank you for the comment.

9. L421, L433: Basin 2 not 6?
We have fixed this. Thank you for the comment.